

# Data assimilation of GNSS Zenith Total Delays from a Nordic processing centre

Magnus Lindskog[1], Martin Ridal[1], Sigurdur Thorsteinsson[2], and Tong Ning[3]

[1]Swedish Meteorological and Hydrological Institute, Norrköping, Sweden.
[2]Icelandic Meteorological Office, Reykjavík, Iceland.
[3]Lantmäteriet, Gävle, Sweden.

*Correspondence to:* Magnus Lindskog (Magnus.Lindskog@smhi.se)

**Abstract.**

Atmospheric moisture-related information obtained from Global Navigation Satellite System (GNSS) observations from ground-based receiver stations of the Nordic GNSS Analysis Centre (NGAA) have been used within a state-of-the-art km-scale numerical weather prediction system. Different processing techniques have been implemented to derive the the moisture-related GNSS information in the form of Zenith Total Delays (ZTD) and these are described and compared. In addition full scale data assimilation and modelling experiments have been carried out to investigate the impact of utilizing moisture related GNSS data from the NGAA processing centre on a numerical weather prediction (NWP) model initial state and on the following forecast quality. The sensitivity of results to aspects of the data processing, observation density, bias-correction and data assimilation have been investigated. Results show a benefit on forecast quality of using GNSS ZTD as an additional observation type. The results also show a sensitivity to thinning distance applied for GNSS ZTD observations but not to modifications to the number of predictors used in the variational bias correction applied. In addition it is demonstrated that the assimilation of GNSS ZTD can benefit from more general data assimilation enhancements and that there is an interaction of GNSS ZTD with other types of observations used in the data assimilation. Future plans include further investigation of optimal thinning distances and application of more advanced data assimilation techniques.

## 1 Introduction

Data assimilation in Numerical Weather Prediction (NWP) optimally blends observations with an atmospheric model in order to obtain the spatial distribution of atmospheric variables and to produce the best possible model initial state. It was early realised that the forecast quality is strongly dependent on an accurate description of the initial state (Lorenz, 1965). There are strong requirements on infrastructure and computing power for today's state-of-the-art high resolution modelling systems. As model resolutions increase it is increasingly important to utilize observations with high spatial and temporal resolution to initialize mesoscale phenomena, such as convective storms and sea breezes.

The meteorological weather services of Sweden, Norway and Finland recently joined forces around a common operational km-scale forecasting system named MetCoOp (Muller et al., 2017). The forecast model used within MetCoOp is developed in the framework of the shared Aire Limite'e Adaptation Dynamique Developpement InterNational (ALADIN)- High-Resolution





Limited-Area Model (HIRLAM) NWP system. This system can be run with different configurations and in MetCoOp the so-called HIRLAM ALADIN Regional Meso-scale Operational NWP In Europe-Application of Research to Operations at Mesoscale (HARMONIE-AROME) is used (Bengtsson et al., 2017). The main components of the ALADIN-HIRLAM NWP system are surface data assimilation, upper-air data assimilation and the forecast model for the forward time integration. The surface data assimilation uses an optimal interpolation scheme (Giard and Bazile, 2000). In the current study a 3-dimensional variational data assimilation (3D-Var) scheme (Fischer et al., 2006) was used for upper-air with a 3 h data assimilation cycle (Seity et al., 2011). The forecast model configuration, e.g. dynamical core and physical parameterizations are described in detail in Seity et al. (2011) and Bengtsson et al. (2017).

The only direct humidity measurements used in the MetCoOp upper-air analysis are provided by vertical profile measurements from radiosondes. In addition, humidity-related information is provided by radar measurements (Ridal and Dahlbom, 2017), by satellite-based information and by moisture-related observations from the Global Navigation Satellite System (GNSS) Zenith Total Delay (ZTD). The disadvantage of all of these observation types, except GNSS ZTD, is that they they are only available at particular times of the day (radiosonde and satellite measurements) or their availability is dependent on weather situation (radar measurements). GNSS ZTD observations, on the other hand, are available at all times with a high temporal resolution (15 minutes), for all weather situations. Moisture-related observations in the form of GNSS ZTD is a relatively new source of mesoscale atmospheric humidity information. ZTD observations obtained from the network of ground-based GNSS receivers contain horizontally dense information on the total columnar amount of water vapour. A number of assimilation studies have shown a positive impact of GNSS ZTD observations on NWP systems at a horizontal model grid resolution of the order of 10 km (De Pondeca and Zou, 2001; Vedel and Huang, 2004; Cucurull et al., 2004; Poli et al., 2007; Macpherson et al., 2008; Yan et al., 2009a, b; Boniface et al., 2009; Benjamin et al., 2010; Shoji et al., 2011; Bennitt and Jupp, 2012; Desroziers et al., 2012). The importance of combining the GNSS data with other types of observations has been highlighted in several studies (Cucurull et al., 2004; Desroziers et al., 2012; Sánchez-Arriola and Navascués , 2007; Sánchez-Arriola et al., 2006). Some first encouraging results from assimilation of these observations at a km-scale horizontal resolution have been obtained (Seity et al., 2011; de Haan, 2013; Sánchez-Arriola et al., 2016) and GNSS ZTD from 28 receiver stations are assimilated operationally in MetCoOp.

The EUMETNET GPS Water Vapour Program (E-GVAP) is a collaborative effort between the European geodetic community and several European national meteorological institutes. The purpose of E-GVAP is to provide atmospheric water vapour observations for use in operational meteorology. ZTD observations obtained from the E-GVAP network of ground-based GNSS receivers contain horizontally dense information and are available with a temporal resolution of up to five minutes and therefore they have the potential to provide humidity related data for km-scale short-range weather forecasting. To stimulate further enhancements in the preprocessing and use of GNSS ZTD observations in NWP and nowcasting applications, in particular when forecasting severe weather, an European COST Action (ES1206) has been ongoing between 2013 and 2017. The action resulted in revitalisation of the Nordic NGAA processing centre, now located at Lantmäteriet in Sweden. It processes GNSS data for a large number of receiver stations, mainly from the Nordic countries. The dense network of GNSS ZTD observations provide an attractive source of supplementary humidity information to the MetCoOp modelling system.



Like all other types of measurements, the GNSS ZTD observations are associated with errors that need to be properly characterized. It has earlier been demonstrated that adaption of variational bias correction (Dee, 2005) to be used together with GNSS ZTD data was successful for handling of systematic observation errors (Sánchez-Arriola et al., 2016). In Sánchez-Arriola et al. (2016) only one predictor was used in the variational bias correction. Earlier, Storto and Randriamampianina (2010) have studied the behaviour of a non-adaptive multi-linear bias correction scheme inspired by the one proposed by Harris and Kelly (2001) and found a benefit of using more than one predictor. The question is whether an adoptive bias correction scheme like the one used by Sánchez-Arriola et al. (2016) would also benefit from using more predictors.

Due to the measurement and processing techniques GNSS observations are very likely to have correlated errors. The difficulties of spatially and temporally correlated observation errors have generally been circumvented in data assimilation by applying thinning of data, or through observation processing algorithms that are assumed to remove the observation error correlations (Stewart et al., 2013). Methods have been developed to account for serially correlated errors (Järvinen et al., 1999) but there is certainly room for improvement regarding spatially correlated errors, although some general research within this area has been carried out (Lin et al., 2000; Liu and Rabier, 2002; Bormann and Bauer, 2010; Stewart et al., 2013). Some studies have focused on GNSS ZTD observations (Kleijer, 2001; Stoew, 2004; Eresmaa and Järvinen, 2005), but the handling of the correlated observation errors is still an active area of research.

GNSS ZTD observations processed by the NGAA centre has been used within the MetCoOp forecasting system, aiming at improving short-range forecasts of in particular moisture, clouds and precipitation. Two different GNSS ZTD processing techniques applied at NGAA are described, compared and evaluated. The sensitivity of the results to various aspects of the GNSS ZTD observation handling and data assimilation is investigated. The evaluation includes both statistcs based on extended parallel experiments and individual case studies.

The paper is organized as follows. The GNSS data processing is the topic of Section 2. In Section 3 the NWP modelling system is described. Section 4 deals with the design of parallel data assimilation experiments and their corresponding results are presented in Section 5. Finally conclusions are presented in Section 6 together with some future plans.

## 2   GNSS data processing

Since June 2016 Lantmäteriet (Swedish Mapping, Cadastre and Land Registration Authority) became one of the analysis centres in E-GVAP and is in charge of the data processing for the GNSS stations in Sweden, Finland, Norway, Denmark and some IGS stations. It includes in total approximately 700 stations. Two near real-time (NRT) ZTD products (NGA1 and NGA2) are currently provided. The NGA1 product is obtained from the Bernese v5.2 (Dach et al., 2007) network solution while NGA2 is given by the GIPSY/OASIS II v.6.2 (Webb and Zumberge, 1993) data processing using the Precise Point Positioning (PPP) strategy (Zumberge et al., 1997). In a network solution there is no need for the precis clock product for the GNSS satellites due to the differential observables. However, the computing time will be exponentially increased as the number of GNSS stations in the data processing increases while the station related errors are correlated to each other. In a PPP processing, each time only the data from one GNSS station is processed meaning that station related errors are independent from others. However, a high



quality of the satellite clock product is critical for the accuracy of a PPP data processing. More details about the two types of data processing can found in Section 2.2 and 2.2.

## 2.1 Post-data processing

In order to obtain the best accuracy on the estimated hourly ZTDs, the coordinates of the stations need to be fixed in a NRT data processing. The fixed coordinates are provided by a post-data processing which is carried out once per day. Due to the latent time of the final orbit products, the post-data processing is taken place for the day two weeks back (14 days). The estimated coordinates will be averaged together with the coordinates estimated for the last six days. The weekly averaged coordinates will be used as the fixed coordinates for a hourly NRT data processing. Although for each station the fixed coordinates are the ones estimated for a day two weeks back, the maximum difference in the height component is less than 1 mm if no significant movements happened on the station, e.g., earthquake, in the last 14 days. Such a small difference will only have an insignificant impact (smaller than 0.3 mm) on the estimated ZTD.

In the post-data processing the acquired GPS phase-delay measurements are used to form ionospheric free linear combinations (LC) that are analysed by Bernese v5.2 using a network solution to estimate station coordinates together with tropospheric parameters. We use the final GPS orbit products provided by CODE ftp.unibe.ch and include an ocean tide loading correction using the FES2004 model (Lyard et al., 2006). The absolute calibration of the Phase Centre Variations (PCV) for all antennas (IGS14.atx) is implemented (Schmid et al., 2007). The tropospheric estimates are updated every two hours while one-hour estimates are given for the station coordinates. A 10° elevation cut-off angle is used and the slant delays are mapped to the zenith using the Vienna Mapping Function 1 (VMF1) (Boehm et al., 2006).

## 2.2 NGA1

The NGA1 product is obtained from a Bernese hourly data processing running in near real-time and using the fixed station coordinates. We use the ultra-rapid GPS orbit products provided by CODE ftp.unibe.ch. The ocean tide loading correction (FES2004) and the antenna PCV absolute calibration are implemented. The tropospheric estimates are updated every 15 minutes and a 10° elevation cut-off angle is used with a Global Mapping Function (GMF) (Boehm et al., 2005). The NGA1 product is currently under the operational status with a time delay of 45 minutes.

## 2.3 NGA2

The NGA2 product is obtained from a GIPSY NRT data processing where the GPS data were analysed by GIPSY-OASIS v6.2 using the PPP strategy with the fixed station coordinates. Currently we use the ultra-rapid GPS orbit and clock products provided by JPL sideshow.jpl.nasa.gov/pub/JPL_GPS_Product/Ultra. The same set-ups are used for the GIPSY data processing, i.e., FES2004 model, antenna PCV absolute calibration, a 10° elevation cutoff angle, and a GMF. The tropospheric estimates are updated every 5 minutes. In addition the single receiver phase ambiguity resolution is also implemented (Bertiger et al.,



2010). The NGA2 product is now under a test mode due to a longer time delay of about 1.5 hours for fetching the JPL ultra-rapid orbit and clock products.

## 2.4 Comparing the datasets

Due to the long time delay in the NGA2 data the NGA1 is the dataset that is sent to E-GVAP for redistribution between member countries. At E-GVAP it is still in test mode but this will change to operational in the near future. Validation is made against a NWP model run carried out at the Royal Netherlands Meteorological Institute (KNMI) together with stations from other processing centres around Europe. The comparison against the model should not be taken as a validation of truth but it makes it possible to compare the results from different processing centres. In Figure 1 an example of such a comparison, taken from http://egvap.dmi.dk/, is shown for the station Onsala in southern Sweden. This is one of the so-called supersites that is included in all processing centres around Europe for comparison. It can be seen that NGA1 compares well with most of the other centers.

In Figure 2 the two solutions from NGAA are also compared against each other. The data are from the reciever in Ballerup (BUDP) just outside Copenhagen. It can be seen that the two solutions are very similar. Interesting to notice is the increased ZTD values around 22-25 June when a major convective storm passed over south western Denmark and the southern part of Sweden. This will be discussed further in the case study in section 5.4. In Figure 3 this time period is zoomed in on and it is easier to see that the two solutions are very similar. The mean difference between NGA1 and NGA2 for June 2016 was 0.65 mm for the Ballerup station and the maximum difference was 21.6 mm. For Onsala the corresponding numbers were 0.26 mm and 17.2 mm respectively.

## 3 The NWP modelling system

The meteorological weather services of Sweden, Norway and Finland share a common operational km-scale forecasting system named MetCoOp (Muller et al., 2017). The forecast model configuration used within MetCoOp is called HARMONIE-AROME (Bengtsson et al., 2017) and is developed in the framework of the shared ALADIN-HIRLAM NWP system. The main components of the ALADIN-HIRLAM NWP system are surface data assimilation, upper-air data assimilation and the forecast model.

The forecast model configuration, e.g. dynamical core and physical parameterizations are described in detail in Seity et al. (2011) and Bengtsson et al. (2017). It has a spectral representation with a non-hydrostatic formulation. Stratiform and deep convective clouds are explicitly represented, while for shallow convection a sub-grid parameterization is applied using the EDMF (Eddy Diffusitivity Mass Flux) scheme. The representation of the turbulence in the planetary boundary layer is based on a prognostic Turbulent Kinetic Energy (TKE) equation combined with a diagnostic mixing length (Cuxart et al., 2000). The radiative transfer of the short-wave spectrum is described with six spectral bands (Fouquart and Bonnel, 1980) and the long-wave radiation is modeled in accordance with Mlawer et al. (1997). Surface processes are modeled using SURFEX (Masson et al., 2013). Lateral boundary conditions were provided by hourly ECMWF operational forecasts. In addition, a




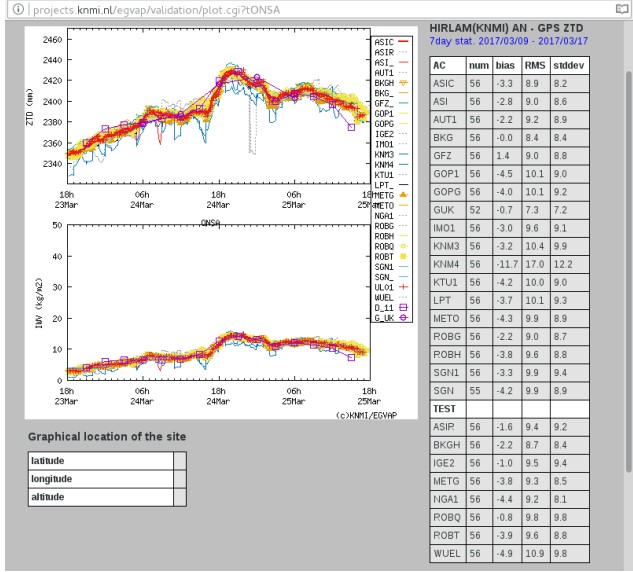

**Figure 1.** Example of validation of ZTD (upper panel) and integrated water vapour (lower panel) from a week in March 2017 for station Onsala, Sweden. Statistics for different processing centres compared to a NWP run is shown in the Table. The data are taken from the E-GVAP web-site.

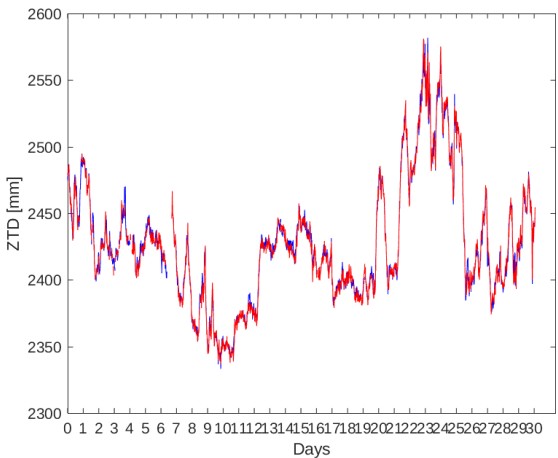

**Figure 2.** Time series of ZTD for June 2016 for the NGA1 (blue) and NGA2 (red) solutions. The data are from the Ballerup (Copenhagen) station in Denmark. The x-axis shows the days in June while the y-axis shows the ZTD in mm.

spectral large scale mixing of the background state, the 3 h HARMONIE forecast, fields with the lateral boundary ECMWF fields was applied. In this way we hoped to benefit from the high-quality large scale information from the ECMWF global forecasts in the regional MetCoOp data assimilation.





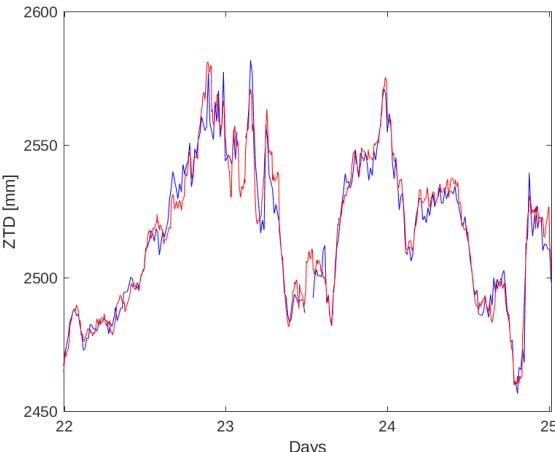

**Figure 3.** Time series of ZTD for 22 to 25 of June 2016 for the NGA1 (blue) and NGA2 (red) solutions. The data are from the Ballerup (Copenhagen) station in Denmark. The x-axis shows the days in June while the y-axis shows the ZTD in mm.

In the MetCoOp setup there are 750 × 960 horizontal grid-points at each of the 65 vertical levels extending up to 10 hPa, which approximately corresponds to a height of 32 kms in the atmosphere. The horizontal grid distance is 2.5 km. The model domain is illustrated by the black frames in Figure 4. In the surface data assi milation synop observations of temperature and relative humidity at the vertical level of two meters were utilized. In addition sea surface temperature and sea ice concentration from an oceanographic model is used. In the MetCoOp upper-air data assimilations conventional types of in-situ measurements were used and these include radiosonde, pilot-balloon wind, SYNOP, ship, and aircraft measurements. In addition radiances from the AMSU-A, AMSU-B/MHS and IASI instruments onboard polar-orbiting satellites are used, as well as surface winds from the Advanced Scatterometer (ASCAT) instrument. Furthermore humidity observations from networks of ground based weather radars and GNSS receiver stations were used. The observations used were obtained from the Global Telecommunications System (GTS), the EUMETSAT Advanced Retransmission Service (EARS), the advanced weather radar network for the Baltic Sea Region (BALTRAD) data hub and the E-GVAP retransmission service.

The surface data assimilation uses an optimal interpolation scheme  (Giard and Bazile, 2000). In the current study a 3-dimensional variational upper-air data assimilation (3D-Var) scheme  (Fischer et al., 2006) was applied within a 3 h data assimilation cycle  (Seity et al., 2011). The climatological background error statistics used in the current study were derived from an ensemble of MetCoOp forecast differences obtained through downscaling of the European Centre for Medium-Range Weather Prediction (ECMWF) Ensemble Data Assimilation (EDA)-based forecast fields. The ECMWF EDA-based forecast fields were horizontally and vertically interpolated to the HARMONIE AROME 2.5 configuration geometry and used as initial conditions for high-resolution nonhydrostatic model runs. The ECMWF EDA uses a T399 horizontal resolution and 91 vertical levels. Then the evolved high-resolution ensemble was scaled to be consistent with the amplitude of the 3 h forecast error for HARMONIE-AROME. The values applied correspond roughly to a GNSS ZTD background-error standard deviation. Recently





ECMWF have increased the horizontal resolution of the EDA system to T639 and demonstrated clear improvements from this change of resolution (Holm et al., 2016). One could expect that re-derivation of the MetCoOp forecast differences utilizing the enhanced ECMWF EDA system would lead to improved MetCoOp background error statistics and thus an improved data assimilation system. We have therefore re-calculated the background error statistics utilizing the enhanced ECMWF EDA system and carried out sensitivity experiments with the new background error statistics. Results are presented in section 5 as an example of how GNSS ZTD data assimilation can gain from more general data assimilation improvements. The background error statistics are specified in terms of assimilation control variable vorticity, unbalanced divergence, unbalanced temperature, unbalanced specific humidity and unbalanced surface pressure (Derber and Bouttier, 1999; Berre, 2000). Important upper-air data assimilation observation handling components are the modelling of observation counterparts with an observation operator, quality control, thinning, bias correction, error specification. The observation operator projects the model state onto the GNSS ZTD observation. Since a variational framework is used, non-linear as well as the corresponding tangent-linear and adjoint versions of the observation operator are needed. The ZTD observation operator ($H$), given a station location (including altitude), calculates the model-equivalent of the GNSS ZTD by integrating the model-calculated refractivity vertically from the station height to the model top, as described in Poli et al. (2007). In the MetCoOp system, following the ideas of Vedel et al. (2001), we have extended the observation operator with the possibility of accounting for the contribution to the ZTD by the part of the atmosphere above the model top. Details of the observation handling within the data assimilation with emphasis on GNSS ZTD is given in Sánchez-Arriola et al. (2016).

The GNSS ZTD observation errors of the observations accepted for the data assimilation were assumed to have a Gaussian error distribution with an observation error standard deviation of 12 mm. This observation error standard deviation was derived from observation minus background and observation minus analysis departures, and it was empirically adjusted so that roughly the same weight was given to the observation and to the background. Objective methods such as the one proposed by Desroziers et al. (2005) could in future be tried instead to tune the observation error variance.

There is also an additional quality control within the assimilation. The purpose of this quality control is to remove observations affected by gross errors and a central part is the background check. The observation, $y_i$, is rejected if it does not satisfy the following inequality:

$$([H(\mathbf{x}^b)]_i - y_i)^2/\sigma_{b,i}^2 > L \times \lambda, \tag{1}$$

where $\lambda = 1 + \sigma_{o,i}^2/\sigma_{b,i}^2$, $L$ is the rejection limit and $[H(\mathbf{x}^b)]_i$ denotes the projection of the model state on observation $i$. In the background-guess check, the background- and observation-error standard deviation were set to 10 mm and 12 mm, consistent with the values used in 3D-Var. The rejection limit for GNSS ZTD observations in the HARMONIE system is currently set to 4. This value results in a relatively strict background quality control of GNSS ZTD observations.

To elevate the effects on the initial state of spatially correlated observation errors caused by for example orographic effects, we apply a spatial thinning of GNSS ZTD observations. The default thinning distance is of the order of 80-100 km. The thinning distance was used when selection receiver stations so that receiver stations closer to each other than 80-100 km are not





used. Also this thinning distance is rather empirically determined. A study of the sensitivity of reducing the thinning distance can be found in section 5. A next step would be to apply objective methods such as the one proposed by Bormann and Bauer (2010) and instead of tuning the thinning distance the observation error covariance could be modelled.

Systematic errors in the GNSS ZTD data is handled by a adaptive variational bias correction (VarBC). Within VarBC the bias is represented by coefficients for the selected predictors. these predictors are estimated within the variational data assimilation process simultaneously as deriving the assimilation control vector for the model state and minimizing the cost function (Dee and Uppala, 2009; Sánchez-Arriola et al., 2016). The bias correction is carried out individually for each receiver station and in the default version only one predictor, in the form of an offset value, is used. However there is also a possibility to introduce more predictors, like 1000-300 hPa thickness, and Total Columnar Water vapour (TCWV). The sensitivity to introducing extra predictors is investigated in section 5.

## 4 Experimental design

In order to investigate the potential benefit in the MetCoOp system of utilizing NGAA GNSS ZTD a number of parallel data assimilation and forecast experiments have been carried out. Furthermore the parallel experiments aimed at investigating the sensitivity of the GNSS ZTD data assimilation to various aspects of the data assimilation. A copy of the MetCoOp operational configuration was run with a 3 h data-assimilation cycle for the period 1-30 June 2016 and with a one-month spin-up period before that. This particular month was chosen because it was characterized by several heavy precipitation events. We expect that additional moisture-related observations should be particularly beneficial for prediction of such weather situations. We ran short-range forecasts every 3 h to provide the background for the next analysis, and we launched forecasts up to 36 h 4 times per day, from 00, 06, 12 and 18 UTC. In total there were four data assimilation studies (**A-D**), each involving two or more parallel data assimilation experiments. These parallel experiments are abbreviated as follows:

**A** Assessing the impact of assimilating GNSS ZTD from the NGAA processing centre.

    **1** Observation usage as in MetCoOp operational, including GNSS ZTD from ROBH, and METO processing centres (CRL).

    **2** Observation usage as in MetCoOp operational, except that GNSS ZTD observation usage was extended to include also receiver stations from the NGAA processing centre, processed with the Bernese approach. (NGA1).

    **3** Observation usage as in MetCoOp operational, except that GNSS ZTD observation usage was extended to include also receiver stations from the NGAA processing centre, processed with the GIPSY approach. (NGA2).

**B** Assessing the impact of different VarBC predictor choices.

    **1** Observation usage as in **A2** above, i.e. utilizing one predictor in the form of an offset value for the GNSS ZTD variational bias correction.





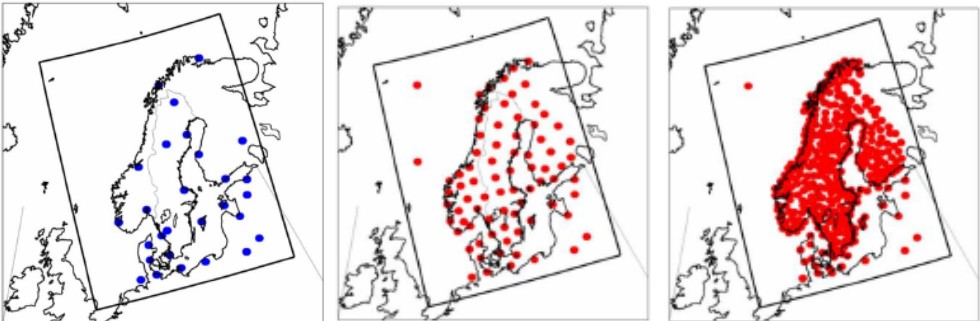

**Figure 4.** MetCoOp model domain (black frame) and GNSS ZTD observation usage for operational MetCoOp (left), NGAA usage with 80-100 km thinning distance (middle) and NGAA usage with 40 km thinning distance (right).

> **2** Observation usage as in **A2** above, except that the variational bias correction was extended to two predictors:offset value and 1000-300 hPa thickness.

> **3** Observation usage as in **A2** above, except that the variational bias correction was extended to two predictors:offset value and Total Columnar Water vapour (TCWV).

**C** Assessing the impact of modifying thinning distances for GNSS ZTD.

> **1** Observation usage as in **A2**, i.e. utilizing one predictor in the form of an offset value for the GNSS ZTD variational bias correction and a GNSS ZTD thinning distance of order of 100 km.

> **2** Observation usage as in **A2**, except that a GNSS ZTD thinning distance of order of 40 km was used.

**D** Assessing the potential benefit of general data assimilation improvements on GNSS ZTD utilization for NWP.

> **1** Observation usage as in **A2**, i.e. utilizing one predictor in the form of an offset value for the GNSS ZTD variational bias correction and a GNSS ZTD thinning distance of order of 100 km.

> **2** Observation usage as in **A2**, except that an improved B matrix was used. is used.

The MetCoOp model domain and the GNSS ZTD observation usage in the operational set-up and in the NGAA ZTD observation usage when applying different thinning distances are illustrated in Figure 4 where the left panel shows case **A1**, the middle panel **A2** (or **C1**) and the right panel shows case **C2**.

## 5  Results

### 5.1  Verification methods

To evaluate the relative quality of the analyses and subsequent forecasts from the different parallel experiments, we verified them against radiosonde and SYNOP observations within the model domain. The verification was carried out for weather





parameters at the surface level and for the upper-air parameters, wind, temperature, and humidity. The model data used in the statistics were the analyses and forecasts of up to 24 h. Special emphasis was put on verification of humidity and precipitation. In addition we used the degrees of freedom for signal (DFS) to study the relative impact of observations in the assimilation system  (Chapnik et al., 2006). DFS is the derivative of the analysis increments in observation space with respect to the observations used in the analysis system. As proposed by  Chapnik et al. (2006) DFS can be computed through a randomization technique:

$$DFS = \frac{\partial H\mathbf{x^b}}{\partial y} \approx (\tilde{\mathbf{y}} - \mathbf{y})R^{-1}(\mathbf{H}\tilde{\mathbf{x}}^a - \mathbf{H}\mathbf{x}^b) - (\mathbf{H}\mathbf{x}^a - \mathbf{H}\mathbf{x}^b), \qquad (2)$$

where $\mathbf{y}$ is the vector of the observations, $\tilde{\mathbf{y}}$ is the vector of perturbed observations, $\mathbf{R}$ is the observation-error covariance matrix, $\mathbf{H}$ is the tangent-linear observation operator for each observation type, $\mathbf{x}^a$ and $\mathbf{x}^b$ are the analysis and the background state, respectively, and $\tilde{\mathbf{x}}^a$ is the analysis produced with perturbed observations. The previous formulation can be applied to any subset of observations  (Randriamampianina et al., 2011). The absolute DFS represent the information brought into the analyses by the different observation types, in terms of amount, distribution, instrumental accuracy and observation operator definition. They offer an insight to the actual weight given to the observations within the analysis system in terms of self-sensitivity of the observations (i.e. sensitivity at location of observation). However, they do not provide any information on the spatial- or cross-correlations between the observations and the analysis.

The different kinds of objective statistical verifications described above were also complemented with a more subjective verification for an individual case study.

## 5.2   Impact on analyses

For the DFS computation the following eight times and dates were chosen four days apart to reduce the interdependency between the initial conditions, and to obtain data from data assimilation cycles covering different times of the day: 0000 UTC (2 June), 0300 UTC (6 June), 0600 UTC (10 June), 0900 UTC (14 June), 1200 UTC (18 June), 1500 UTC (22 June), 1800 UTC (26 June) and 2100 UTC (30 June).

In Figure 5 the DFS calculated separately for different observation types and parameters are shown. The values represent the sum over the observations belonging to the same subset of Eq. 2 calculated for each individual observation. Results are shown for the four experiments **A1**, **A2**, **C2** and **D2**. The rest of the experiments all have DFS similar to **A2** and are therefore not shown. Comparing the DFS of **A1**, **A2** and **C2** shows that the contribution from GNZZ ZTD increases with an increasing number of GNSS ZTD observations. One can also notice a clear interaction with moisture-related observations from IASI and radar. The larger DFS of GNSS ZTD after increasing the number of GNSS observations was associated with an increase in DFS from radar-based humidities and a decrease in DFS from the IASI instrument, providing satellite-based humidity information. It is also evident by comparing **A2** and **D2** that by improving the background error statistics one can increase the DFS for GNSS ZTD and also of other observations. From DFS scores not shown, the impact on analysis from NGA1 and NGA2 was



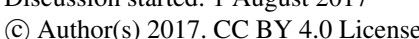

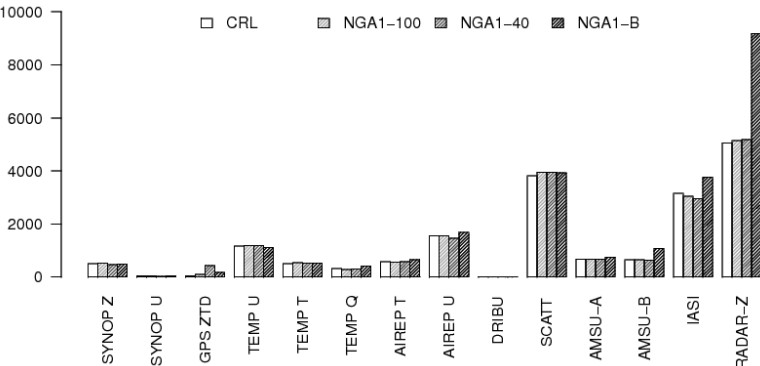

**Figure 5.** Degree of Freedom of Signal sub-divided into various observation types for the four experiments **A1**, **A2**, **C2** and **D2**. Results were based on data from eight different data assimilation cycles.

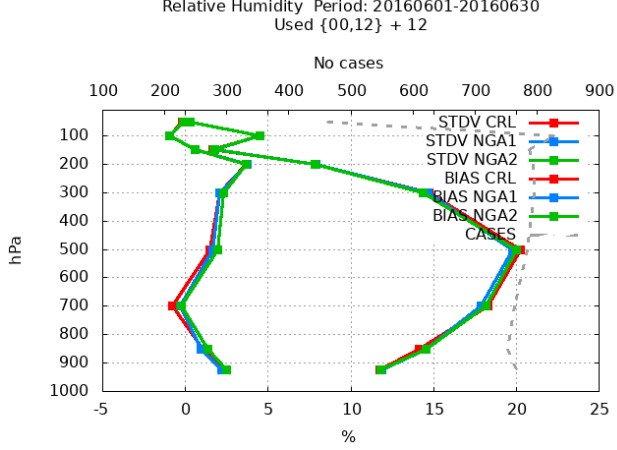

**Figure 6.** Bias and standard deviation of +12 h relative humidity (unit: %) forecasts as function of vertical level for verification against radiosonde observations. Scores are for experiments **A1** (red), **A2** (blue) and **A3** (green).

very small and the impact on DFS for GNSS ZTD of introducing more predictors in the variational bias correction of GNSS ZTD was also very limited.

## 5.3    Statistical verification of forecasts

In Figure 6 the scores for verification of +12 hour and +24 hour relative humidity forecasts of the experiments **A1**- **A3** against radiosonde observations within the domain are shown for different vertical levels. A small positive impact on forecasts can be seen from utilizing NGAA ZTD observations. The positive impact was slightly more pronounced when the NGAA observations were in the form of NGA1. For forecasted variables other than humidity, the observed impact was small.





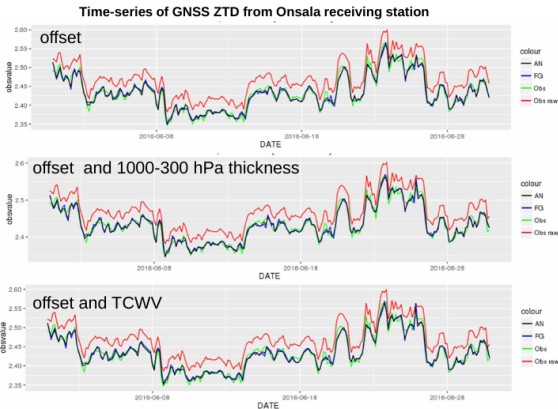

**Figure 7.** One-month time-series of GNSS ZTD (unit: m) from the Onsala receiver station. Analysed (black), background (blue), observation after bias correction (green) and observation before bias correction (red).

The impact of utilizing two predictors in the the variational bias correction of GNSS ZTD is small, not only in terms of DFS. As another example, Figure 7 shows, for one particular receiver station (Onsala), a one-month time-series during the experiment GNSS ZTD of background state equivalent (FG), analysis (AN), observed value before bias correction (OBS RAW), observed value after bias correction (OBS), for the three different experiments **B1**- **B3**. It can be seen that the bias correction is properly working, managing to correct for the systematic difference between the raw observation and the model state equivalents. On the other hand, it was evident that the time-evolution of bias-corrected observation was very similar between the three different runs. The difference between introducing the second predictor in the form of 1000-300 hPa thickness or TCWV was very small.

The small impact of introducing additional predictors in the adaptive bias correction was also confirmed by forecast verification scores. Figure 8 illustrates the impact on +12 and +24 h relative humidity forecasts, for verification against radiosonde observations. As for forecasts of other variables (not shown), the impact was small.

The sensitivity of modifying the thinning distance applied to GNSS ZTD observations is illustrated in Figure 9. From the left part of the Figure it can be seen that in terms of standard deviation the impact was rather small except for improved humidity forecasts at the lowest levels when reducing the thinning distance from 80-100 km to 40 km. The right part of the Figure shows that this improvement was present at forecast ranges up to 36 hours. In terms of bias, on the other hand one can see from the left Figure that there was an increased positive humidity bias throughout the lower troposphere when reducing the thinning distance. Again, for forecasts of other variables (not shown) the impact is small. An increased humidity bias when reducing the thinning distance was noticed also by Sánchez-Arriola et al. (2016) and it was speculated whether the lack of high resolution complementary unbiased humidity information of nearby GNSS ZTD receiver stations affect each other during the spin-up phase of predictor coefficients. In Sánchez-Arriola et al. (2016) only conventional types of observations were used in addition to the GNSS ZTD observations. Our study confirms that the increased bias when reducing the thinning distance was present also when a substantial amount of humidity related remote sensing observations, such as AMSU-B/MHS, IASI





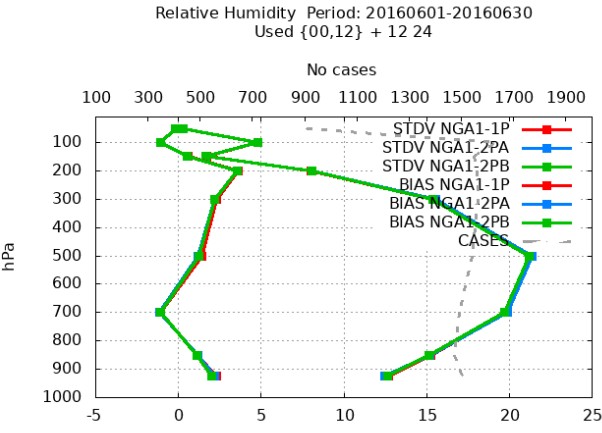

**Figure 8.** Bias and standard deviation of +12 and +24 h relative humidity (unit: %) forecasts as function of vertical level for verification against radiosonde observations. Scores are for experiments **B1** (red), **B2** (blue) and **B3** (green).

and radar derived humidities were assimilated in addition to GNSS ZTD. It should be kept in mind however, that none of these data sources are assumed to be bias free. For AMSU-B/MHS and IASI a variational bias correction is applied and for radar derived moisture information a pre-processing utilizing the model background field is applied through a method described in detail in Caumont et al. (2010) and Wattrelot et al. (2014). Our results from section 5.2 hint that there is a relation between IASI, radar and GNSS ZTD impact when modifying the GNSS ZTD thinning distance. However, the interaction of reduction of thinning distances and increased bias needs to be better understood before one can fully benefit from reducing the GNSS ZTD thinning distance. This is one of the aims for future in-depth studies with the MetCoOp data assimilation system.

In addition to improvement of the low level humidity forecasts when reducing the thinning distance to 40 km one can see a slight improvement in forecasts of cloud cover and more pronounced improvements in precipitation forecasts, as illustrated in Figure 10, in terms of Kuiper skill score. Thus, despite the increased bias in humidity related to the reduction in thinning distance, the improvements in terms of standard deviations for humidity forecasts resulted also in improvement in the humidity-related variables cloud and precipitation. The question whether improvements also can be seen in an individual case is addressed in section 5.4.

When investigating the improvements to the system that can be brought by adding new observations and by refinements of the observation handling it is also useful to get an idea of how much the extraction of information from the new observations, as well as from all the other observations, can be improved by general data assimilation improvements. In our case, the general data assimilation improvements were given by an improved representation of background error statistics. The improved background error statistics had an positive impact on the forecasts, shown in Figure 11 for temperature and relative humidity scores. A positive impact was found also on surface pressure forecasts and wind forecasts (not shown).





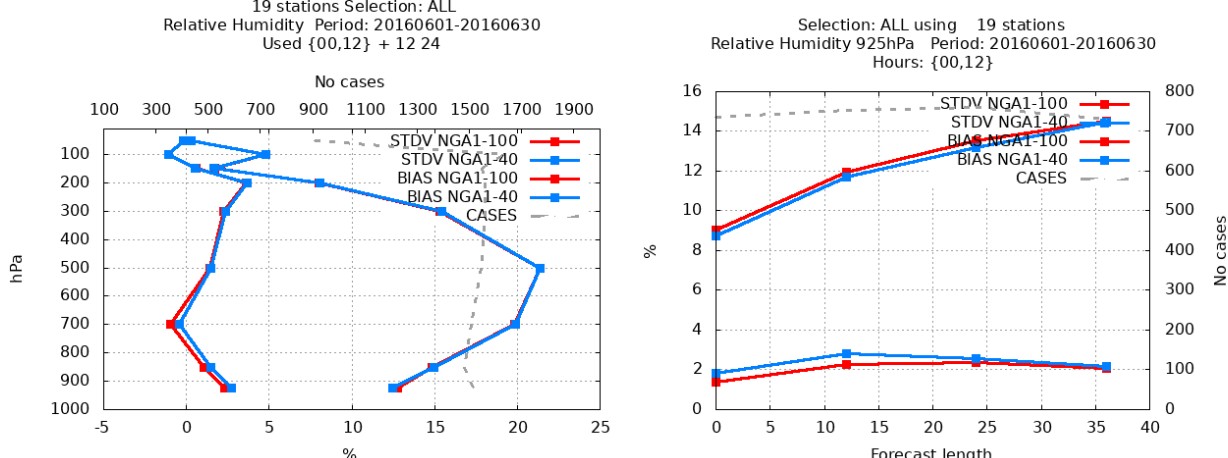

**Figure 9.** Bias and standard deviation of +12 and +24 h relative humidity (unit: %) forecasts as function of vertical level for verification against radiosonde observations. Scores are for experiments **C1** (red) and **C2** (blue).

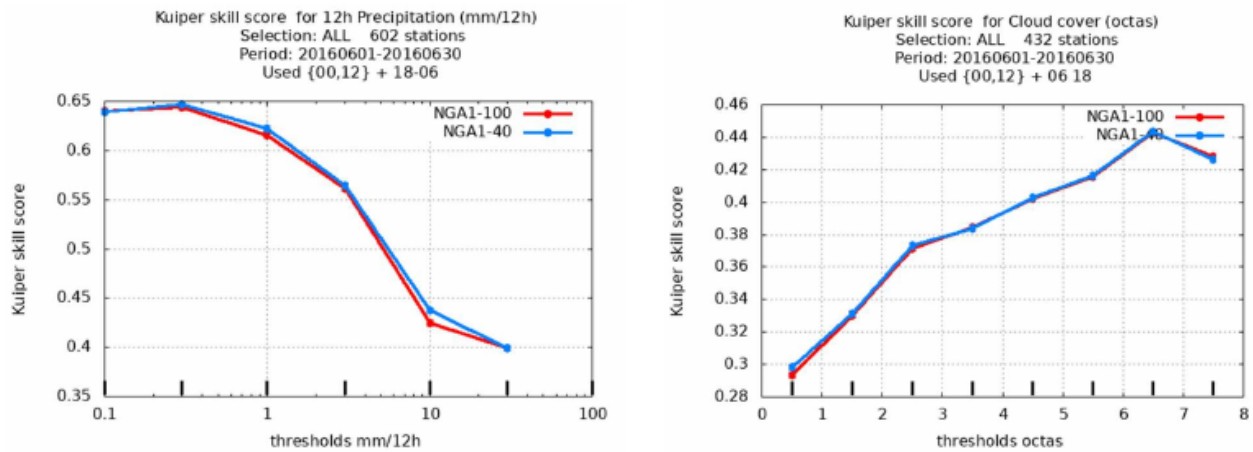

**Figure 10.** Kuipper skill score for 12 h accumulated precipitation (left) and +6 and +18 h cloud forecasts (right) for verification against synop stations in the domain.

General data assimilation improvements, like the improved B matrix presented here, influenced more aspects and observations of the data assimilation system than just GNSS ZTD observations. It is important to keep in mind that such general improvements can also be supportive in obtaining more useful information from both newly introduced observation types as well as the already exisiting ones that have been in use for some time.



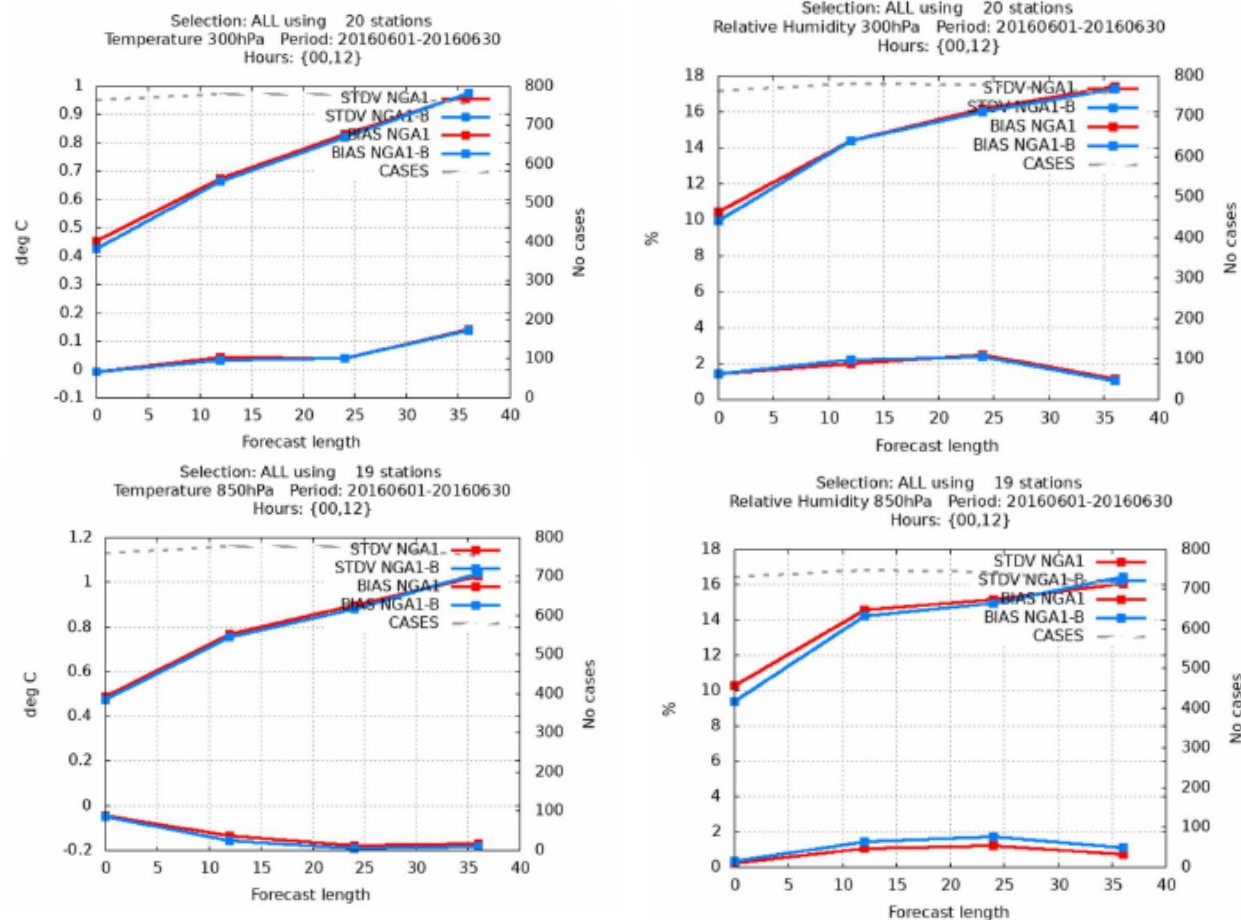

**Figure 11.** Bias and standard deviation of temperature (unit: K) and relative humidity (unit: %) as function of forecast range. Scores for temperature (left column) and relative humidity (right column) at the vertical levels of 300 hpa (upper) and 850 hPa (lower). Scores are for experiments **D1** (red) and **D2**.

## 5.4 Case study

To investigate whether the modification of thinning distance have any noticeable effect on individual weather situations we investigated one particular case in more detail. This individual case selected was a heavy precipitation case that took place over south-western Denmark and the southern part of Sweden during the night/early morning between 24-25 of June 2016. The upper row of Figure 12 shows the radar derived precipitation rate at 02.00 local Swedish time (00.00 UTC) and at 05.00 (03.00 UTC). The middle and lower panels show the corresponding precipitation forecasts for the runs with 80-100 km and 40 km thinning distance, respectively. At 02.00 Swedish local time the forecasts of the two runs were rather similar, but as the system moved toward the north-east more of the intensity and structure in accordance with observations was retained in the run with 40 km thinning distance. Note that for this particular case for both forecasts of **C1** and **C2** there was a phase error





since the precipitation system is located further eastward in radar-based observations than in the corresponding forecasts, and in addition the accumulation intervals differed between observation and forecasts. Nevertheless, forecasts of **C2** were in better accordance with radar based observations than the **C1** ones.

## 6 Conclusions

The processing of GNSS ZTD data from the newly vitalised NGAA processing centre has been described in detail. It is shown that these data have the capability to enhance the NWP forecasts, in particular for humidity when introduced, in addition to other observations, in the HARMONIE-AROME model. The sensitivity of the forecasts to the two solutions of estimating ZTD provided and to various settings in the GNSS ZTD data processing has been investigated. The two different methods of estimating ZTD generated very similar results and the impact on the forecasts was therefore also very small. It was also found that the results were rather insensitive to the number of predictors used in the variational bias control. In this study only two predictors were tested at the same time. It might be useful as a next step to test more than two and also try other paramaters, e.g. surface pressure. Opposite to the small impact from the VarBC predictors the results were rather sensitive to the choice of thinning distance applied. There are potential improvements from reducing the thinning distance of the ZTD observations to make use of more data, but there are also related issues. Reducing the thinning distance resulted in increased humidity forecast biases in the lower troposphere. This may have been due to increased influence from correlation errors and needs to be investigated further to find the best trade-off between the number of observations and the influences of error correlations. In general the horizontal observation error correlations need to be investigated further, for example by applying techniques proposed by  Bormann and Bauer (2010) and in the yet further step modelled correlations.

The assimilation of GNSS ZTD in NWP can benefit from more general data assimilation improvements, such as enhanced description of statistical information or improved data assimilation algorithms. In this paper this was highlighted by providing an example in the form of an additional run carried out with what we think is an enhanced description of the background error statistics. Clearly the enhanced description resulted in better use of the GNSS ZTD observations in the NWP system. It is important however, to keep in mind that such general data assimilation aspects not only influence the GNSS ZTD observation usage but also all other observations. In addition, further developments of the data assimilation algorithms, e.g. the impact on utilization of GNSS ZTD observation in a 4-dimensional variational data assimilation, will be investigated.

*Acknowledgements.* The authors would like to thank Jan Johansson at Chalmers Technical University for support with revitalising the NGAA processing centre. The help from Dr Roger Randriamampianina with computations of DFS is also greatly appreciated. The work was carried out within the framework of the European COST Action ES1206 concerned with Advanced Global Navigation Satellite Systems tropospheric products for monitoring severe weather events and climate.



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







**Figure 12.** Radar based accumulated precipitation (upper row, unit mm/h) and associated forecasted accumulated precipitation (unit: mm/3h) based on **C1** (middle row )and **C2** (lower row). Left column is for accumulation period starting at 20260625 00 UTC and right column is for accumulation period starting at 20260625 03 UTC. The forecasts are initiated from 20260624 12 UTC.