# Peer review of "Data assimilation of GNSS Zenith Total Delays from a Nordic processing centre"

_Atmospheric Chemistry and Physics, 2017_

## Referee Comment (RC1) · Anonymous Referee #1 · 8 Aug 2017

**Review of acp-2017-567: Data assmilation of GNSS Zenith Total Delays from a Nordic processing centre, by Lindskog, Ridal, Thorsteinsson and Ning.**

**General**
This is a nice article assessing the impact of GNSS ZTDs from the Nordic countries (except Iceland) on the quality of forecasts produced with the Harmonie-Arome NWP model. Several strategies for ZTD data thinning and variational bias correction are tested. In addition two types of GNSS ZTD estimation is tested.

The manuscript is easy to read, and can be accepted with minor corrections and addons.

**Specific**
Be consistent in defining and then using abreviations. For example you define NWP twice, and the first time it is not when you first write numerical weather prediction.

There's a tendency to cite some papers very often. Try to limit yourself to only cite a given paper once in one paragraph.

In constrast to most other data assimilated in NWP, ZTD is a parameter that improves with time; as satellite orbits, clock errors etc. become better determined, moving from being based on predictions to being based on observations. For this reason it is good to talk about ZTD being an estimate, not an observation.

p 2 Either here, or in the section on the NWP model setup, provide at short, conceptual explanation how radar rain rate/reflectivity is turned into humidity.

p 2 Consider refering also to the recent review by Guerova et al (the GNSS4SWEC review paper).

p 3 top of page, and again p 9: How do you know the biases handled with VarBC are biases of the GNSS ZTDs? NWP also have biases. In addtion observation operators might include short cuts, for example regarding the corrections for the offset between model orography and GNSS antenna, which will also show manifest themselves as biases.

p 3 4. line from bottum. Precis -> Precise

p 5, section 2.4 : On the intercomparison of nga1 and nga2. Why not compare also to ZTD estimates obtained in GNSS post processing, serving as "truth".

p 5+, section 3: Regarding the NWP setup there is too much sharing of whole sentences between the introduction and section 3. Reduce the amount of duplication.

p 5 last line: I presume you use 6 or 12 hourly ECMWF forecasts with one hour time resolution as boundaries?

p 7 3. line: assi milation -> assimialation

p 8 ".. bias correction, error specification" -> "..bias correction, and error specification"

p8 You use a fixed sigma_O for all sites and seasons, and discuss using Desroziers method to determine

sigma_O instead. Have you considered also looking at the O-B distributions and give preference to those GNSS sites that are "most Guassian" in this regard?
In this connection, is the data thinning a random thing, or is it always ZTDs from the same sites that are assimilated?

p 8, 3. last line: To elevate -> To alleviate

p 10.  Please provide a few words about the DFS signal per observation.

p 15, figure 9.   Use markers for 6 or 12 hours, to be consistent with your NWP cycling and verification frequency.

p 15, figure 10. Notice that automated cloud cover measurements from some instruments have systematic problems at the very highest and lowest values.

p 16, figure 11. Use markers for 6 or 12 hours.

p 16. The selection of the case study is a bit unfortunate. The area with strong precipitation is quite close to the boundary of your NWP area, and the atmospheric flow in the period is from South-West toward North-East, ie. into your area near the main precip. In addition you have for some reason not included GNSS ZTDs from the E-GVAP Dutch, German, and Polish processing centers, which would have provided plenty of additional ZTDs for assimilation. As heavy local precip in Southern Sweden and Denmark is often related to humid air arriving from South-West, these are probably important observations to include in your operational NWP.

Figure 12. I strongly recommend to provide maps of observed 3 hour precipitation for comparision to the NWP 3 hour precip. maps. Either raingauge adjusted radar precip, or raingauge based precip. The instantanuous radar rainrate images are misleading. Alternatively the NWP precip should be for a much shorter period.
For the type of heavy precipitation 12 h to 18 h is a long forecast, and you might excuse yourself and provide observed versus modelled precip with an offset in time. You have already mentioned a phase error in the text. Looking at synop precip data in our database it appears to me the precip your NWP dumps between 0 and 3 UTC actually fell between 19 and 23 UTC the day before.

Figure  12. The year is 2016, not 2026.

---

## Referee Comment (RC2) · Anonymous Referee #2 · 11 Aug 2017

**Review of acp-2017-567:** *Data assimilation of GNSS Zenith Total Delays from a Nordic processing centre,* **by Magnus Lindskog, Martin Ridal, Sigurdur Thorsteinsson, and Tong Ning**

**General Comments**

This paper is reasonably well written and I find its topic quite interesting in the context of limited-area mesoscale model Data Assimilation.

Anyway, I feel there are a few issues that might deserve a closer look, as discussed in the detailed comments below.

**Specific Comments**

Introduction

- A so extended explanation of the model used could not be so necessary here due to it is done later in Section 3.

- When is says "and GNSS ZTD from 28 receiver stations are assimilated operationally in MetCoOp" could be explained a little more.

Section2:

-The writting (wording) of this section should be revised.

Section4:

-More clarification of the different experiments should be made, for example when talking about A , B or D experiments comment that they are all using 100km thinning, when the experiments are from A, C or D they are using a constant offset, etc. Maybe a table with all the experiments and properties could help.

-Why do you specify the experiment name just on A1, A2, and A3 cases ?

Section5:

-The Case study Section was clearly explained but may be not enough to show the improvement of using one thinning distance or other. Supporting those graphics with the use of  SAL verification method for example could result in a  more complete description of the chosen case study.

**Technical Corrections**

Introduction

-There are several repeated "they"

-Statistics instead of statistcs

Section2:

-NGAA name should be mentioned at the beginning.

-ZTD instead ZTDs

-NRT: explanation is needed when it is used for the first time

2.2. The title could be changed to NGA1 product/ dataset, not just NGA1

2.3 The same

2.4 Please review the wording. Receiver instead of reciever

Section3:

-assimilation instead of assi  milation

-data assimilation instead of data assimilations

-…control variable vorticity, unbalanced divergence, …  review that sentence

-alleviate instead of elevate

-Last paragraph: review the writing

Section5:

5.2- Review the first sentence writing

5.3.-

    -H+12 and H+24 instead of +12 and +24

    -Fig 6: H+12 instead of +12

    -"For forecasted variables other than humidity…", review this sentence

    -".. in the form of NGA1...", it is not a' form' but a product or dataset.

    -Review the writing of the last part of the section

    -Fig 8, 9 and 10: H+12 and H+24 instead of +12 and +24

5.4- Figure 12: 2016 instead 2026

---

## Author Comment (AC1) · 5 Sep 2017

**Authors Response
to
Reviewer I comments
on
manuscript
acp-2017-567
Data assmilation of GNSS Zenith Total Delays from a Nordic processing centre
by
Lindskog, Ridal, Thorsteinsson and Ning**

**September 2017**

Dear Reviewer I,
We are very thankful for quick and excellent comments. We
have taken action to all suggestions and comments. See our
response below in blue. The comments and suggestions were
well in accordance with the ones made by a separate
Reviewer. We have updated the paper based on the feedback
and we feel that we have benefitted by the comments by
Reviewer I. The modifications will appear in the revised
paper to be submitted shortly after 26 September. The
constructive comments and suggestions made by the two
anonymous Reviewers were greatly appreciated and we have
made a note of that in the Acknowledgements of the revised
manuscript.
Best Regards
Magnus, Martin, Sigurdur and Tong

**Reviewer I comments and responses**

**General**
This is a nice article assessing the impact of GNSS ZTDs from the Nordic countries
(except Iceland) on the quality of forecasts produced with the Harmonie-Arome NWP
model. Several strategies for ZTD data thinning and variational bias correction are tested.
In addition two types of GNSS ZTD estimation is tested. The manuscript is easy to read,
and can be accepted with minor corrections and addons.
Thank you, we are happy to have been notified that Reviewer I is, overall, positive to the
content and form of our manuscript.

**Specific**
Be consistent in defining and then using abreviations. For example you define NWP
twice, and the first time it is not when you first write numerical weather prediction.
There's a tendency to cite some papers very often. Try to limit yourself to only cite a
given paper once in one paragraph.

As suggested, we have reduced the number of citations for a given paper to only once in a paragraph. We think that we are also now consistent in defining and using abbreviations. Also Reviewer II pointed out that we omitted to define at least one abbreviation. We now only define abreviations once, the first time it appears. The exception is that when abbreviations appear both in the abstract and in the text we define it twice. The reason is that we want both the abstract itself and the text without the abstract to be self-contained. We have tried to avoid abreviations in the abstract to make it easily readable but GNSS and ZTD appear many times and are important and central to the manuscript. Therefore we defined these abreviations also in the abstract.

In constrast to most other data assimilated in NWP, ZTD is a parameter that improves with time; as satellite orbits, clock errors etc. become better determined, moving from being based on predictions to being based on observations. For this reason it is good to talk about ZTD being an estimate, not an observation.

*This is true. For simplicity however we have chosen to refer to it as an observation throughout the manuscript. This is now also stated in the introduction.*

p 2 Either here, or in the section on the NWP model setup, provide at short, conceptual explanation how radar rain rate/reflectivity is turned into humidity.
*As suggested, we have in the NWP model setup added a short explanation of how radar rain rate/reflectivity is turned into humidity and given a reference to a more detailed description.*

p 2 Consider refering also to the recent review by Guerova et al (the GNSS4SWEC review paper).
Good suggestion. We have now mentioned the work by Guerova *et al.* in the introduction and added a reference.

p 3 top of page, and again p 9: How do you know the biases handled with VarBC are biases of the GNSS ZTDs? NWP also have biases. In addtion observation operators might include short cuts, for example regarding the corrections for the offset between model orography and GNSS antenna, which will also show manifest themselves as biases.
We agree that something more should be written regarding the possible causes of systematic differences between GNSS ZTD biases. We have therefore, as suggested, added the following sentences in the manuscript:
'The sources of bias in the ZTD observation data with respect to the ZTD model data may be due to several reasons, such as GNSS data-processing algorithms (use of mapping functions, formulation of hydrostatic delay, errors in the conversion of ray path to zenith delay) and systematic errors in both the model fields and the ZTD observation operator. In particular, a low model top will generally result in a systematically too-low model equivalent of the GNSS ZTD observations.'

p 3 4. line from bottum. Precis -> Precise

Corrected.

p 5, section 2.4 : On the intercomparison of nga1 and nga2. Why not compare also to ZTD estimates obtained in GNSS post processing, serving as "truth".

Figures 2 and 3 have been replotted where nga1 and nga2 were compared to the ZTDs obtained from GNSS post processing.

p 5+, section 3: Regarding the NWP setup there is too much sharing of whole sentences between the introduction and section 3. Reduce the amount of duplication.
We totally agree and related comments came also from another Reviewer. We have therefore removed several sentences concerned with NWP setup details from the introduction and also slightly modified the NWP setup part to reduce duplicated sentences.

p 5 last line: I presume you use 6 or 12 hourly ECMWF forecasts with one hour time resolution as boundaries?
Good point. We have clarified by rewriting the sentence as: 'Lateral boundary conditions were provided by 6 hourly ECMWF operational forecasts with one hour time resolution.'

p 7 3. line: assi milation -> assimialation
'assi milation' corrected to 'assimilation'.

p 8 ".. bias correction, error specification" -> "..bias correction, and error specification"
*Thanks. We have now changed from '.. bias correction, error specification' to '..bias correction and error specification'.*

p8 You use a fixed sigma_O for all sites and seasons, and discuss using Desroziers method to determine sigma_O instead. Have you considered also looking at the O-B distributions and give preference to those GNSS sites that are "most Guassian" in this regard? In this connection, is the data thinning a random thing, or is it always ZTDs from the same sites that are assimilated?
Thank you for a relevant comment pointing out the need for further clarification of the thinning procedure. The approach suggested by the Reviewer is more or less the one we have applied. We added the following sentences to the manuscript:
'The thinning is applied in the form of a static whitelist based on studies of data availability and of observation minus background equivalent statistics from a spin-up period. Thus the thinning is static so that for each data assimilation cycle of observations from the  same set of GNSS ZTD receiver stations are used.'

p 8, 3. last line: To elevate -> To alleviate
Corrected.

p 10. Please provide a few words about the DFS signal per observation.
As suggested we have provided a few words about the DFS signal per observation by adding the following sentences to the manuscript: 'There is also the possibility of

estimating the DFS per observation through calculation of relative DFS, by normalizing the absolute DFS by the amount of the observations belonging to a specific subset. Here we have, however, chosen to focus on absolute DFS.'

p 15, figure 9. Use markers for 6 or 12 hours, to be consistent with your NWP cycling and verification frequency.
As suggested, we have changed to use markers for 12 hours, to be consistent with verification frequency.

p 15, figure 10. Notice that automated cloud cover measurements from some instruments have systematic problems at the very highest and lowest values.
Based on this comment, we have now added a sentence, including two references to the revised manuscript: 'It should, however, be kept in mind that there are some known problems related to precipitation and cloud measurements Rodda and Dixon 2012; Wagner and Kleiss, 2016}.'

p 16, figure 11. Use markers for 6 or 12 hours.
As suggested, we have changed to use markers for 12 hours.

p 16. The selection of the case study is a bit unfortunate. The area with strong precipitation is quite close to the boundary of your NWP area, and the atmospheric flow in the period is from South-West toward North-East, ie. into your area near the main precip. In addition you have for some reason not included GNSS ZTDs from the E-GVAP Dutch, German, and Polish processing centers, which would have provided plenty of additional ZTDs for assimilation. As heavy local precip in Southern Sweden and Denmark is often related to humid air arriving from South-West, these are probably important observations to include in your operational NWP.
We do not think that the particular case we have chosen occurs so close to the lateral boundary. We agree that there might nevertheless be some influence from the lateral boundaries. We furthermore agree that we would in this case probably have benefitted from GNSS ZTD observations from Dutch, German, and Polish processing centres, as well as radar-based humidity information from Danish and Dutch radars. However, our point is that this is the NWP setup used operationally and forecasts are made for Southern Sweden also during this type of weather situation. We would like to see whether our newly added GNSS ZTD from the NGAA processing centre had the potential to improve the forecast, and the relative merits of the different configurations, like 40 km thinning distance. From Figure 4 it can be seen that with 40 km thinning distance there are several GNSS ZTD observations in the area of relevance for the case study. Including even more observations and processing centres will come as a next step in future research.

Figure 12. I strongly recommend to provide maps of observed 3 hour precipitation for comparision to the NWP 3 hour precip. maps. Either raingauge adjusted radar precip, or raingauge based precip. The instantanuous radar rainrate images are misleading. Alternatively the NWP precip should be for a much shorter period. For the type of heavy precipitation 12 h to 18 h is a long forecast, and you might excuse yourself and provide observed versus modelled precip with an offset in time. You have already mentioned a

phase error in the text. Looking at synop precip data in our database it appears to me the precip your NWP dumps between 0 and 3 UTC actually fell between 19 and 23 UTC the day before.

This is a good suggestion. We have included a new Figure showing 3 h accumulated precipitation from rain gauges that registered rain during the particular case. As suggested, due to the phase error, we have chosen to shift the accumulation period for one hour as compared with the accumulation in the forecasts.  We believe that this Figure has strengthened the illustration of the case study and thank the Reviewer for his suggestion. We have also added a text in the case study section related to the newly added Figure.

Figure 12. The year is 2016, not 2026.

Corrected.

---

## Author Comment (AC2) · 5 Sep 2017

**Authors Response**
**to**
**Reviewer II comments**
**on**
**manuscript**
**acp-2017-567**
**Data assmilation of GNSS Zenith Total Delays from a Nordic processing centre**
**by**
**Lindskog, Ridal, Thorsteinsson and Ning**

**September 2017**

Dear Reviewer II,
We are very thankful for quick and excellent comments. We have taken action to all suggestions and comments. See our responses below in blue. The comments and suggestions were well in accordance with the ones made by a separate Reviewer. We have updated the paper based on the feedback and we feel that we have benefitted from the comments by Reviewer I. The modifications will appear in the revised paper to be submitted shortly after 26 September. The constructive comments and suggestions made by the two anonymous Reviewers were greatly appreciated and we have made a note of that in the Acknowledgements of the revised manuscript.
Best Regards
Magnus, Martin, Sigurdur and Tong

<h1 style="text-align:center">Reviewer II comments and responses</h1>

**General Comments**

This paper is reasonably well written and I find its topic quite interesting in the context of limited-area mesoscale model Data Assimilation.

Anyway, I feel there are a few issues that might deserve a closer look, as discussed in the detailed comments below.

Thank you for your generally positive view on the manuscript and for your constructive comments that are well in line with the comments of Reviewer I. We have revised our text, and suggestions and detailed responses can be found below.

**Specific Comments**

Introduction

- A so extended explanation of the model used could not be so necessary here due to it is done later in Section 3.

We totally agree and related comments came also from Reviewer II. We have therefore removed several sentences concerned with NWP setup details and made sure these are instead covered in the NWP setup section.

- When is says "and GNSS ZTD from 28 receiver stations are assimilated operationally in MetCoOp" could be explained a little more.

Good suggestion. We have added the following text to explain a bit more about the 28 receiver stations that were assimilated operationally into the MetCoOp system:

'These 28 receiver stations have been selected from the rather few receiver stations over the MetCoOp domain. Often these are supersites, processed by several centres for comparison purposes. MetCoOp operationally uses data processed by the Met Office in the United Kingdom and by the Royal Observatory of Belgium.'

Section2:

-The writting (wording) of this section should be revised.

Thank you, we have revised the section.

Section4:

-More clarification of the different experiments should be made, for example when talking about A , B or D experiments comment that they are all using 100km thinning, when the experiments are from A, C or D they are using a constant offset, etc. Maybe a table with all the experiments and properties could help.

This is an excellent suggestion. We have added a number of sentences describing what is in common between experiments in different studies. However, in our opinion, after adding these sentences, it is not necessary to add an additional Table to the manuscript describing the experiments. The newly added sentences are: ' Note that in all experiments of studies A,  C, and  D only one predictor in the form of an offset value was used. In all experiments in studies B and  D a 100 km thinning distance was used. All experiments in studies A, B and C used the operationally used B matrix and all experiments in studies B, C and  D used the NGA1 data set, processed with the Bernese approach.'

-Why do you specify the experiment name just on A1, A2, and A3 cases ?
Thanks for pointing this out. It was a mistake and it was inconsistent. We have chosen to remove the specification of experiment names also from A1-A3, since we think these are not needed.

Section5:
-The Case study Section was clearly explained but may be not enough to show the improvement of using one thinning distance or other. Supporting those graphics with the use of SAL verification method for example could result in a more complete description of the chosen case study.
We appreciate this comment and also another Reviewer had similar thoughts that supporting graphics would be beneficial. We therefore included one additional Figure in the manuscript showing gauge-adjusted precipitation. Due to the phase error, we have chosen to shift the accumulation period for one hour as compared with the accumulation in the forecasts.  We believe that this Figure has improved the illustration of the case study and thank the Reviewer for his suggestion. We have also added text in the case study section related to the newly added Figure. Finally we have modified the radar Figure to show gauge-adjusted accumulated precipitation, rather than instantaneous precipitation as before.  Regarding SAL verification, we think that it would be mainly beneficial for an extended period and not just for the one case. Based on that and the fact that we already have several different verification approaches applied in the manuscript we chose not also to include SAL verification

**Technical Corrections**

Introduction

-There are several repeated "they"
Thanks, at one point 'they they' was replaced by 'they' and at another point we simply removed 'they', since it was not needed. Now 'they' appears only once or twice in the section.

-Statistics instead of statistcs
Corrected.

Section2:

-NGAA name should be mentioned at the beginning.
It is now mentioned at the beginning of section 2.

-ZTD instead ZTDs
Modified as suggested.

-NRT: explanation is needed when it is used for the first time
Abbreviation is defined the first time it is used. One sentence has been added in order to give a brief explanation about what near real time means in E-GVAP.

2.2. The title could be changed to NGA1 product/ dataset, not just NGA1
As suggested, we have modified the title from 'NGA 1' to 'NGA 1 dataset'.

2.3 The same
As suggested, we have modified the title from 'NGA 2' to 'NGA 2 dataset'.

2.4 Please review the wording. Receiver instead of reciever
The section has been partly re-written and 'Reciever' is corrected to 'receiver'.

Section3:

-assimilation instead of assi milation
Corrected.

-data assimilation instead of data assimilations
Corrected.

-...control variable vorticity, unbalanced divergence, ... review that sentence
We have now modified the sentence and split it into two sentences, to make it easier to read.

-alleviate instead of elevate
Corrected.

-Last paragraph: review the writing
Thanks, we have now reviewed and improved the wording in the last paragraph of section 3.

Section5:

5.2- Review the first sentence writing
Thanks, we have now modified the sentence and split it into two sentences, to make it easier to read.

5.3.-
-H+12 and H+24 instead of +12 and +24
Thanks for making us aware of inconsistency between text here and in Figures 6, 8, 9 and 10. However we prefer to write the forecast ranges as '+12 and +24 h'.

-Fig 6: H+12 instead of +12
*Thanks for pointing this out, but again we prefer to write the forecast ranges as '+12 and +24 h'.*

-"For forecasted variables other than humidity...", review this sentence
The sentence has been modified to be clearer.

-".. in the form of NGA1...", it is not a' form' but a product or dataset.
Modified as suggested.

-Review the writing of the last part of the section
As suggested we have reviewed the writing of the last part of the section now modified at some places to make the text clearer and easier to read.

-Fig 8, 9 and 10: H+12 and H+24 instead of +12 and +24
*Thanks for pointing this out but again we prefer to write the forecast ranges as '+12 and +24 h'.*

5.4- Figure 12: 2016 instead 2026
Corrected.